



# Interrelationships among soil nitrogen transformation rates, functional gene abundance and soil properties in a tropical forest with exogenous N inputs

5    Yanxia Nie [1], Xiaoge Han[1], Jie Chen[2], Mengcen Wang[3], Weijun Shen [1]

[1]Key Laboratory of Vegetation Restoration and Management of Degraded Ecosystems, South China Botanical Garden, Chinese Academy of Sciences, Guangzhou 510650, PR China

[2]Research Institute of Tropical Forestry, Chinese Academy of Forestry, Guangzhou 510520, PR China

10   [3] Ministry of Agriculture Key Laboratory of Molecular Biology of Crop Pathogens and Insects, Zhejiang University, Hangzhou 310058, PR China

*Correspondence to*: Weijun Shen (shenweij@scbg.ac.cn)



**Abstract.** Elevated nitrogen (N) deposition affects soil N transformations in the N-rich soil of tropical forests. However, the change in soil functional microorganisms responsible for soil N cycling remains largely unknown. Here, we investigated the variation in soil inorganic N content, net N mineralization ($R_m$), net nitrification ($R_n$), inorganic N leaching ($R_l$), $N_2O$ efflux and N-related functional gene abundance in tropical forest soil over a two-year period with four levels of N addition. The responses of soil N transformations (*in situ* $R_m$, $R_n$ and $R_l$) to N additions were delayed during the first year of N inputs. The $R_m$, $R_n$, and $R_l$ increased with the medium nitrogen (MN) and high nitrogen (HN) treatments relative to the control treatments in the second year of N additions. Furthermore, the $R_m$, $R_n$, and $R_l$ were higher in the wet season than in the dry season. The $R_m$ and $R_n$ were predominately driven by the lower C:N ratio under N addition in the dry season but by higher microbial biomass in the wet season. Throughout the study period, high N additions increased the annual $N_2O$ emissions by 78%. Overall, N additions significantly facilitated soil N availability ($R_m$ and $R_n$) and N loss ($R_l$ and $N_2O$ emission), which had a stimulating effect on N transformations. In addition, the MN and HN treatments increased the ammonia-oxidizing archaea (AOA) abundance by 17.3% and 7.5%, respectively. Meanwhile, the HN addition significantly increased the abundance of *nirK*-denitrifiers but significantly decreased the abundance of ammonia-oxidizing bacteria (AOB) and *nosZ*-containing $N_2O$ reducers. To some extent, the variation in functional gene abundance was related to the corresponding N transformation processes. Partial least squares path modelling (PLS-PM) indicated that inorganic N contents had significant negative direct effects on the abundances of N-related functional genes in the wet season, implying that chronic N deposition would have a negative effect on the N-cycling-related microbes and the function of N transformation.





## 1   Introduction

Due to anthropogenic activity in recent decades, the increased atmospheric reactive nitrogen (N) deposition in terrestrial ecosystems has altered the N status and dynamics (Galloway et al., 2008). Excessive N inputs to forest ecosystems will certainly influence soil N cycling and ecosystem function. In the last three decades, several studies have focused on the impacts of N deposition on soil N cycling in northern and temperate forests (Aber et al., 1989;1998;Gundersen et al., 1998;Nave et al., 2009;Tian et al., 2018). However, in recent years, tropical forests have received the most dramatic increases in N deposition and are considered as N-rich areas (Hietz et al., 2011;Liu et al., 2013). In southern China, forest ecosystems, such as the hotspots of N deposition receiving 13.8-113 kg N ha$^{-1}$ year$^{-1}$ through precipitation, have reached N saturation status (Fang et al., 2008;Chen et al., 2016;Yu et al., 2018). Little is known about the hazards of constant N inputs on N-saturated forest ecosystem functioning. More attention should be focused on examining the effects of N addition on soil N transformations in N-rich tropical forests.

Soil N availability and turnover are quantified by the N transformation rates in the forest soil (Gao et al., 2016;Patel and Fernandez, 2018). Few previous studies have reported the alteration of N transformation rates after N additions, but these studies have had inconsistent results due to the different soil types, soil ages, N status and duration of N additions in tropical forest ecosystems (Lohse and Matson, 2005;Corre et al., 2010;Chen et al., 2016). Lohse and Matson (2005) reported that first-time and long-term N additions did not change the rate of net N mineralization ($R_m$) but significantly increased the net nitrification rate ($R_n$) and delayed the nitrate leaching rate in a 4.1-million-year-old N-rich and phosphorus (P)-limited forest soil. Corre et al. (2010) documented that the difference in soil transformation rates between the control and N-addition plots was apparent only after chronic (9-year) N additions in a lowland forest but was obvious with short-term (1-year) N additions in a montane forest. Chen et al. (2016) found that a 6-year N addition significantly increased $N_2O$ emission and nitrate leaching but decreased $R_m$ and $R_n$ in a tropical broadleaf forest, possibly due to the alteration of the soil microbial community composition and reduction of enzyme activity with N addition. In contrast, significant increases in $R_m$, $R_n$ and $R_l$ were observed with a 3-year N addition in the adjacent broadleaf forest (Fang et al., 2011). Until now, only a small number of studies have directly quantified soil N transformation rates in tropical forests, and the mechanisms of their conflicting responses to N additions are still unclear (Cheng et al., 2019).

A global meta-analysis showed that N deposition had a negative effect on soil microbial growth, diversity, composition, and function (Zhang et al., 2018c;Wang et al., 2018a), but soil N mineralization was mainly driven by soil microorganisms (Ollivier et al., 2011;Li et al., 2019b). Ammonia oxidation, the first and rate-limiting step of autotrophic nitrification, is performed by ammonia-oxidizing archaea (AOA) and ammonia-oxidizing bacteria (AOB) harbouring AOA *amoA* and AOB *amoA* genes, respectively, which are valuable indices for predicting soil potential nitrification rates (Petersen et al., 2012). AOA play



a dominant role in ammonia oxidation in acidic forest soil and have a positive correlation with gross nitrification rates (Isobe et al., 2012). In addition, elevated N deposition enhances N loss by nitrate leaching and denitrification in tropical forest soil (Chen et al., 2016). The second step in denitrification of reducing $NO_2^-$ to nitric oxide is catalysed by copper-containing reductase (encoded by the *nirK* gene) or cytochrome cd1-containing reductase (encoded by the *nirS* gene) (Braker et al., 2000). Previous studies have shown that *nirK* denitrifiers are more sensitive to environmental changes than are *nirS* denitrifiers (Chen et al., 2010;Li et al., 2019a). Furthermore, the abundances of the *nirK* gene are positively related to potential denitrification rates in an acidic forest soil (Zhang et al., 2018b). The reduction of $N_2O$ to $N_2$ catalyzed by nitrous oxide reductase (encoded by the *nosZ* gene) plays a vital role in mitigating $N_2O$ emissions (Liu et al., 2014;Nie et al., 2016). Therefore, the combination of soil N transformation processes and functional gene abundances is essential to better explain the response mechanism of the soil N cycle to N additions, the relationships between the abundances of soil N-related functional genes and N transformation rates, and the effects of N addition on soil N-related functional microbes.

Soil N transformation rates are thought to be primarily controlled by environmental factors, including temperature, precipitation, carbon to nitrogen (C:N) ratio, soil organic matter (SOM) content, tree species, soil texture and pH (Templer et al., 2005;Chen et al., 2017;Ribbons et al., 2018;Song et al., 2018). Importantly, the contents of soil organic carbon (SOC) and carbon to nitrogen ratio (C:N) are the key factors that determine soil N dynamics in terrestrial ecosystems (Templer et al., 2012;Li et al., 2014;Liu et al., 2017;Fujii et al., 2018). On the other hand, N inputs to forests could alter soil properties. For instance, elevated N deposition can result in soil acidification (Lu et al., 2014;Mao et al., 2017) and a relatively lower soil C:N ratio and lower available P in the forest soil (Shi et al., 2018). The tropical forest soil itself is P-limited and acidic; thus, it is essential to assess the complex interactions between soil physiochemical characteristics and N transformation rates under N deposition.

Here, we investigated the effects of N addition on $R_m$, $R_n$, $R_l$, $N_2O$ emission and N-related functional gene abundance within two years using the *in situ* intact soil core incubation method in an acidic tropical forest. We hereby investigate (1) the effects of short-term N addition on N availability (i.e., N mineralization and nitrification) and N loss (i.e., nitrate leaching and $N_2O$ emission) in N-rich tropical forest soil; (2) the correlations between the variation in soil functional microbial abundances and the corresponding N transformation rates; and (3) the seasonal patterns of N transformations with different soil temperatures and moisture in the dry and wet seasons.

## 2    Materials and methods

### 2.1 Study sites

The study was carried out in the Dinghushan Biosphere Reserve (DHSBR) (112°10′ E, 23°10′ N) in
Guangdong Province of southern China. An experiment using a gradient of nitrogen addition was used to
simulate N deposition in an old-growth and highly weathered evergreen broad-leaved forest with a history
of more than 400 years. The climate of this forest is considered a humid monsoon with an annual average
temperature of 21 °C and a mean annual precipitation of 1927 mm (Mo et al., 2006;Zhao et al., 2011).
The minimum monthly mean temperature in this study area is 12.6 °C in January, and the maximum
monthly mean temperature is 28.0°C in July (Mo et al., 2006). The elevation of this site ranges from 300
to 355 m above sea level. The major tree species of the study site are *Castanopsis chinensis*, *Schima
superba*,      *Cryptocarya      chinensis,*      and      *Randia      canthioides*.
In this site, the wet season is concentrated from April to September (approximately 80% of the annual
rainfalls), and the dry season extends from October to March (approximately 20% of the annual rainfalls).
In addition, the soil type in this region is classified as strongly acidic lateritic red earth formed from
sandstone with a pH below 4.0 (Mo et al., 2006;Zhang et al., 2008a).

### 2.2 Experimental design

Four concentrations of $NH_4NO_3$ were applied: control (0), low N (LN, 35 kg N ha$^{-1}$ year$^{-1}$), medium N
(MN, 70 kg N ha$^{-1}$ year$^{-1}$), and high N (HN, 105 kg N ha$^{-1}$ year$^{-1}$). Twelve (4 treatments × 3 replicates)
scattered plots (15 m × 15 m per plot) were randomly distributed in the study site and established in
October 2013; the plots were surrounded by buffer strips (> 10 m wide) to avoid the disturbance of surface
runoff and flow diffusion between adjacent plots. The corresponding dose of N ($NH_4NO_3$) solution (30 L)
and an equal amount of water (without $NH_4NO_3$) were evenly sprayed over the N-treated and control
plots, respectively, below the canopy using a knapsack sprayer (i.e., a low rate of 0.1 L m$^{-2}$ was applied
to avoid liquid effects) at the end of each month starting in September 2014.

### 2.3 Soil N transformations

Soil net mineralization, nitrification and inorganic N leaching rates were determined nine times from
October 2014 to October 2016 using the *in situ* resin-core incubation method (Reichmann et al.,
2013;Chen et al., 2017). In each plot, six soil sampling sites were evenly distributed in uphill and downhill
areas. At each sampling site, a pair of PVC tubes (5 cm in diameter and 17 cm in length) were inserted
into the soil surface layer (10 cm depth) after the surface litter was removed. A resin bag containing 30 g
ion exchange resin (cation exchange resin: anion resin = 1:2) was placed in the bottom of one PVC tube
(accounting for approximately 2 cm of the PVC tube) under a 10 cm soil layer. The resin cores in the PVC
tubes were incubated *in situ* for 30 days in the field prior to the collection of the soil samples and resin
bags to measure the concentrations of soil $NH_4^+$-N and $NO_3^-$-N. The other PVC tube with a 10 cm soil




core was taken immediately, and then the soils in the PVC tubes were mixed thoroughly (six total soil cores in each plot) into a composite soil sample for further analysis. Soil samples were divided into two parts. One part was passed through a 2-mm sieve and used to analyse the initial concentration of soil $NH_4^+$-N and $NO_3^-$-N, and a small part of the fresh soil was kept at -80°C to extract soil DNA for
quantifying the functional microorganisms. The other part was air-dried at room temperature, and then it was passed through a 100-mesh sieve to estimate the basic soil physicochemical properties. Soil net mineralization, nitrification and inorganic N leaching were calculated according to the following formulas:

$$R_m = \frac{(NH_4^+ - N_{i+1} - NH_4^+ - N_i) + (NO_3^- - N_{i+1} - NO_3^- - N_i)}{t_{i+1} - t_i} \tag{1}$$

$$R_n = \frac{NO_3^- - N_{i+1} - NO_3^- - N_i}{t_{i+1} - t_i} \tag{2}$$

$$R_l = \frac{(NH_4^+ - N_{i+1} - l) + (NO_3^- - N_i - l)}{t_{i+1} - t_i} \tag{3}$$

where $t_i$ and $t_{i+1}$ are the beginning and end dates of the incubation period, respectively; $NH_4^+$-$N_i$ and $NH_4^+$-$N_{i+1}$ are the contents of soil $NH_4^+$-N before and after incubation, respectively, and $NO_3^-$-$N_i$ and $NO_3^-$-$N_{i+1}$ are the concentrations of soil $NO_3^-$-N before and after incubation, respectively (Li et al., 2018a). $NH_4^+$-$N_{i+1}$-$l$ and $NO_3^-$-$N_{i+1}$-$l$ are the contents of $NH_4^+$-N and $NO_3^-$-N in the resin, respectively.

In addition, the concentrations of $NH_4^+$-N and $NO_3^-$-N in the resin were used to calculate the ammonium and nitrate leaching rates respectively. Soil $N_2O$ emissions were monitored using the closed chamber method, and $N_2O$ concentrations were measured with a gas chromatograph (Agilent 7890A, Agilent Technologies, USA) as previously described (Chen et al., 2017).

**2.4 Soil physiochemical properties**

The soil organic carbon (SOC) was estimated using the external heating method with potassium dichromate ($K_2Cr_2O_7$). To obtain the total nitrogen (TN) and total phosphorus (TP), semi-micro Kjeldahl digestion and molybdenum antimony colorimetric approaches were performed, respectively. The contents of soil $NH_4^+$-N and $NO_3^-$-N were detected with 1 M KCl extraction by indophenol-blue colorimetry and double wavelength (220 nm and 275 nm), respectively, using a spectrophotometer (UV-6000, China). Soil
pH was measured by a pH metre with a glass electrode (Horiba F-71S, Japan) (soil: water ratio, 1:2.5 dry wt/v). Soil microbial carbon (MBC) and soil microbial nitrogen (MBN) were determined on a TOC analyser (Shimadzu TOC-VCSH Analyser) by the fumigation-extraction method (Vance et al., 1987).
2.5 Quantification of the abundances of soil functional genes
Soil DNA was extracted using a PowerSoil®DNA Isolation Kit (MOBIO Laboratories, Carlsbad, CA,
USA). DNA concentrations were quantified on a Qubit 2.0 fluorometer (Life Technologies, Carlsbad, CA, USA). Subsequently, quantitative PCR was performed on an ABI 7500 CFX96 Optical Real-Time Detection System (Bio-Rad Laboratories, Inc., Hercules, CA) to quantify the abundances of N-cycling functional genes, including AOA *amoA* and AOB *amoA* genes in nitrification and *nirK* and *nosZ* genes in



denitrification. The pair primes of these functional genes are shown in Table S1. The total volume (20 μl) of the reaction systems contained 10 μl SYBR® Premix Ex Taq™ (TaKaRa Biotech, Japan), 0.4 μl forward and 0.4 μl reverse primer, 0.4 μl Rox Reference Dye II (TaKaRa Biotech, Japan), 1 μl amplification template (genomic DNA) and 7.8 μl sterile ddH$_2$O. The preparation of standard curves and the details of the amplification conditions were conducted as described in Table S2. The amplification efficiencies of qPCR ranged from 95.3% to 103.0%, and the R$^2$ values of the calibration curves were ≥ 0.98.

## 2.6 Statistics

One-way analysis of variance (ANOVA) was used to compare the differences in inorganic N concentrations, soil N transformations, and soil functional gene abundances with the least significant difference (LSD) test for multiple comparisons using SPSS (SPSS 18.0, SPSS Inc., Chicago, USA). Redundancy analysis (RDA) was conducted to determine the comprehensive relationships among soil physiochemical properties, functional gene abundance and N transformations using Canoco 5.0 (Wageningen UR, Netherlands). The correlation coefficients of soil properties, soil N transformations and functional genes were calculated using PAST (version 2.16). The Partial least squares path modelling (PLS-PM) was carried out to test the effects of inorganic N, soil conditions, microbial biomass, and functional gene abundance on soil N transformation rates (R$_m$, R$_n$, R$_l$ and N$_2$O emission) using the "plspm" package in R (version 3.3.3).

## 3   Results

### 3.1 Soil properties and inorganic N contents

The soil C:N ratio in this study site ranged from 11.3 to 18.5, and the pH was between 3.7 and 3.9 (Table S3). The HN addition decreased the SOC, C:N ratio and pH by 14.1%, 9.3% and 1.4%, respectively. The soil TN showed no significant difference between the control and N-treated plots after N addition (Table S3). The concentrations of MBC and MBN decreased obviously by 15.1% and 14.5% respectively in the HN treatment plots in the dry season (Table S3). The contents of soil NH$_4$$^+$-N and NO$_3$$^-$-N significantly increased with N addition ($P < 0.05$, Fig. 1). Our results showed that the amounts of soil NH$_4$$^+$-N and NO$_3$$^-$-N in the MN and HN plots were significantly higher than those in the control plots. The mean value of NH$_4$$^+$-N accounted for 25.1% of the mean total inorganic N, and the NH$_4$$^+$-N /NO$_3$$^-$-N ratio ranged from 0.05 to 0.97. Over the entire study period, the mean soil NH$_4$$^+$-N contents in the LN, MN, and HN treatment plots increased by 27.5%, 38.3%, and 38.6%, respectively. Similarly, the mean concentrations of NO$_3$$^-$-N in these three plots increased by 0.4%, 29.3% and 37.2%, respectively.

### 3.2 Soil net N mineralization and nitrification rates

The results showed that *in situ* R$_m$ and R$_n$ significantly increased after one year of N addition in the MN and HN plots ($P < 0.05$, Fig. 2a and b). However, there were no significant differences in both N



transformation rates between the control and N-treated plots during the first N-treated year ($P > 0.05$). The range of *in situ* $R_m$ (from 4.9 to 44.9 mg N kg$^{-1}$ month$^{-1}$) over the first year of N addition was obviously lower by approximately 50% than the range (from 10.0 to 108.6 mg N kg$^{-1}$ month$^{-1}$) over the second year of N addition. In addition, the responses of $R_m$ and $R_n$ to N addition exhibited different seasonal patterns.

The mean values of $R_m$ in the LN, MN and HN plots in the wet season were 60.3%, 18.5%, and 50.2% higher than those in the dry season over the second year of N addition, respectively. Similarly, the mean value of $R_n$ in the wet season in these three N-treated plots was 1.5-, 1.2-, and 1.3-fold higher than those in the dry season within the same period of N addition.

### 3.3 Inorganic N leaching and N₂O emission

The HN addition significantly increased the ammonium-leaching rates (Fig. 3a), but the ammonium leaching rates accounted for only a small proportion (less than 20%) in the total of $R_l$ and were found to range from 0.08 to 10 mg N kg$^{-1}$ month$^{-1}$. After a one-year period of N additions, the nitrate leaching rates significantly increased in the MN and HN treatment plots ($P < 0.05$, Fig. 3b). The $R_l$ was significantly correlated with the nitrate leaching rate (Fig. 3d, $R = 0.939$, $P < 0.001$), indicating that inorganic N

leaching was predominantly determined by nitrate leaching. The mean values of $R_l$ in the LN, MN, and HN treated plots in the wet season were 1.22, 0.56, and 1.11 times greater than those in the dry season, respectively (Fig. 3e). The addition of N significantly increased the annual N₂O emission (Fig. 3f, $P < 0.05$), showing increases of 18.3%, 18.4% and 77.7% in the LN, MN and HN, respectively, in comparison to the control plots. In addition, a strong positive correlation was observed between the soil $NO_3^-$-N

concentration and nitrate leaching rate in the wet season ($R = 0.63$, $P < 0.001$) (Table 1b). This finding suggested that the accumulation of $NO_3^-$-N content with N addition might accelerate N loss from the acidic forest soil.

### 3.4 Soil microbial functional genes

As shown in Fig. 4a, the copy numbers of the archaeal AOA *amoA* gene ranged from $1.7 \times 10^8$ to $5.2$

$\times 10^8$ g$^{-1}$ dry soil. Although AOA abundance showed no significant difference in all treatments, its mean value increased by 17.3% and 7.5% in the MN and HN plots, respectively, compared with the value in the control plots. AOA abundance showed a significantly negative correlation with soil pH ($R = -0.64$, $P < 0.01$) and a positive correlation with $NO_3^-$-N content ($R = 0.47$, $P < 0.05$) in the dry season (Table 1a). However, the MN and HN additions significantly decreased the copy numbers of the AOB *amoA* gene ($P$

$< 0.05$ and $P < 0.01$, respectively, Fig. 4b). In addition, AOA were more abundant than AOB in the acidic forest soils. The ratio of AOA:AOB abundance ranged from 9.5 to 191.2. However, the abundance of *nirK* genes significantly increased in the second year of HN addition ($P < 0.01$, Fig. 4c). Initially, the abundance of *nosZ* genes decreased in the HN-treated plots compared with that in the control plots in January 2015 and January 2016 ($P = 0.057$, Fig.4d). However, the differences between both were

weakened with the duration of N addition.



## 3.5 Interactions among N transformation rates, soil physicochemical properties and functional gene abundance

RDA was carried out to separately determine the relationship between the soil biotic/abiotic factors and N transformation rates for the dry season and wet seasons. RDA in the dry season was confirmed as unreliable because the $P$ value of the RDA was >0.05 (data not shown). Linear correlation analysis showed that the C:N ratio had significant negative correlations with both the $R_m$ and $R_n$ ($R = -0.45$, $P < 0.05$, Table 1a) but had positive relationships with the abundance of AOA $amoA$, AOB $amoA$ and $nosZ$ genes ($R = 0.44$, $P < 0.05$; $R = 0.58$, $P < 0.01$; $R = 0.48$, $P < 0.05$, respectively). In addition, no significant correlations were found between N transformation rates and biotic factors in the dry season. In the wet season, the first two axes of the RDA explained 65.3% of the total variance in all determined biotic and abiotic parameters and N transformation rates of the soil samples (Fig. 5). The $R_m$, $R_n$ and $R_l$ had significantly positive correlations with the soil $NO_3^-$-N contents, MBN, MBC, SWC, SOC and TN. In contrast, the above N transformation rates had significantly negative relationships with soil pH and $NH_4^+$-N contents. Similarly, $N_2O$ emission was significantly positively correlated with the MBN, MBC, SWC, SOC and TN but significantly negatively correlated with the soil $NH_4^+$-N content. According to the above analysis, we found more complex relationships among the biotic and abiotic factors and N transformations in the wet season than in the dry season.

The PLS-PM was constructed to integrate the complex interrelationships among environmental factors, microbial biomass and soil N transformations in the wet season (Fig. 6). The results showed that inorganic N had positive direct effects on soil conditions (path coefficient = 0.78, $P < 0.001$), microbial biomass (path coefficient = 0.11, $P > 0.05$) and N transformations (path coefficient = 0.18, $P > 0.05$). However, inorganic N had a negative direct effect on N-related functional gene abundance (path coefficient = -0.7, $P < 0.01$). Soil conditions had a positive direct effect on microbial biomass (path coefficient = 0.75, $P < 0.001$). The positive direct contributors to N transformations were inorganic N (path coefficient = 0.18, $P > 0.05$) and microbial biomass (path coefficient = 0.44, $P > 0.05$). In contrast, the negative direct effects on N transformations were soil conditions (path coefficient = -0.07, $P > 0.05$) and N-related functional gene abundance (path coefficient = -0.37, $P = 0.09$).

## 4   Discussion

### 4.1 Effects of N addition on N transformation rates

In contrast to N-limited temperate forests, the N-rich tropical broadleaved forest soil in the DHSBR was considered to be N saturated (Fang et al., 2008). In our study, no significant differences of $R_m$, $R_n$ and $R_l$ were found in the control and N-treated plots during the first-year of N addition, which was possibly ascribed to plant uptake of mineral N and soil N retention (Fang et al., 2011;Gurmesa et al., 2016). However, $R_m$, $R_n$ and $R_l$ significantly increased in the MN and HN treatment plots after one year of N

addition (Fig. 2a, b and Fig. 3c). This result is in agreement with the hypothesis early proposed that once N input exceeds the total biotic demands, it will form a status of N saturation and subsequently promote N mineralization, nitrification, N loss through nitrate ($NO_3^-$-N) leaching and $N_2O$ emissions in boreal and temperate forest ecosystems (Aber et al., 1989;1998). The strong increments of $R_m$, $R_n$ and $R_l$ under N additions lasted only for a short term (from October 2015 to July 2016, Fig.2a, b and Fig.3c). Therefore, our study in the tropical forest provides evidence of the stimulating effects of N inputs on N transformation processes (i.e., *in situ* $R_m$, $R_n$ and $R_l$).

The significant increases in $R_m$ and $R_n$ in the second year of N addition are consistent with the result of the previous study showing significant increases of the two N transformation processes after a 3-year N addition (Fang et al., 2011). However, significant decreases in $R_m$ and $R_n$ after a 6-year N addition were previously demonstrated in the adjacent tropical forest (Chen et al., 2016). The different effects of short-term and long-term N addition on $R_m$ and $R_n$ are possibly caused by the reasons below. First, long-term N additions could lead to high amounts of $NO_3^-$-N accumulation relative to short-term N additions, which may form high osmotic potential and ion toxicity and directly affect soil microorganisms (Wang et al., 2018a). Second, long-term N addition results in a lower soil pH (Lu et al., 2014) and an increase in $Al^{3+}$ content which is toxic to soil microorganisms (He et al., 2012). Third, long-term N deposition has negative impacts on protein depolymerization (Chen et al., 2018), which is considered a rate-limiting step of organic N mineralization (Jan et al., 2009;Mooshammer et al., 2014).

Although the rates of nitrate leaching measured under the 10 cm soil layer might overestimate the N loss attributed to plant uptake below this layer, the result is in agreement with the previous studies of substantial nitrate leaching under N deposition (Fang et al., 2009;Chen et al., 2016). The inorganic N leaching ($R_l$) mainly resulted from nitrate leaching (Fig. 3a, b, c, and d). This result is because it is easier to lose the negatively charged $NO_3^-$-N from the soil while $NH_4^+$-N tends to be taken by plants in acidic forest soils (Fang et al., 2011;Chen et al., 2017). Hall and Matson (1999) found that $N_2O$ emissions were higher in the P-limited tropical forest than in the N-limited forest. In this study, the mean rates of annual soil $N_2O$ emissions in the control plots were $40.4 \pm 6.5$ and $45.1 \pm 5.7$ µg $N_2O$–N m$^{-2}$ h$^{-1}$ in the first two years after N additions, respectively, which were obviously higher than the results of $29.3 \pm 1.6$ µg $N_2O$–N m$^{-2}$ h$^{-1}$ reported by Zhang et al. (2008a), implying that increasing N deposition should exist in natural forest ecosystems. In addition, the rates of $N_2O$ emissions ($95.0 \pm 9.0$ µg $N_2O$–N m$^{-2}$ h$^{-1}$) in the HN treatment plots were significantly higher than those in the LN, MN and control plots, indicating that the soil $N_2O$ emission flux was dependent on the N-addition gradients (Zhang et al., 2008a;Fang et al., 2011;Chen et al., 2016).

## 4.2 Responses of the abundances of microbial functional genes to N additions

AOA play a more important role than do AOB in ammonia oxidation of acidic soils (Zhang et al., 2012;Tang et al., 2016). Similar to the results of a previous study, AOA were more abundant in the acidic forest soil (Isobe et al., 2012). The abundance of AOA slightly increased with 10 months of N addition,

but no significant decrease was found in all treatments (Fig. 4a). On the other hand, the abundance of AOB significantly decreased in the MN- and HN-treated plots compared to that in the control plots (Fig. 4b). A previous study partially confirmed that a 6-year N input increased AOA abundance but decreased AOB abundance in an acidic subtropical forest soil (Shi et al., 2018). The reason for these phenomena might be (1) the decreasing soil pH in the tropical forest soil with elevated N deposition (Lu et al., 2014). AOA could adapt well to strongly acidic soil conditions but AOB tended to be higher in neutral or slightly alkaline soils over all terrestrial ecosystems (Nicol et al., 2008;Hu et al., 2013;Wang et al., 2019) and were more sensitive to N enrichment (Ning et al., 2015). In addition, some previous studies reported that the AOA and AOB ratio increased with decreasing soil pH (He et al., 2007;Shen et al., 2008;Yao et al., 2011;Tang et al., 2019), and (2) the lower $NH_3$ availability with decreasing soil pH, which may have limited the direct substrate for nitrifiers, and AOA are more competitive than are AOB (He et al., 2012;Shi et al., 2018). A previous study documented that a 9-year N addition had no influence on soil N-related functional gene copy numbers in a temperate steppe (Zhang et al., 2018a). These differences may be due to the different soil types and N statuses (N-limited or N-rich) in different ecosystems.

The abundances of *nirK*-denitrifiers are positively related to potential denitrification (Zhang et al., 2018b;Tang et al., 2019). Here, we found that HN additions initially decreased *nirK* gene abundance in the first year of N addition but significantly increased *nirK* gene abundance in the second year of N addition (Fig. 2b), which is possibly ascribed to the accumulation of soil $NO_3^-$-N and the subsequent acceleration of denitrification. The HN addition decreased *nosZ* gene abundance in the earlier stage of N addition, but a decrease in the difference was also observed with the duration of N addition. These variations in *nirK* and *nosZ* gene abundance and the greater abundance of *nirK* than the *nosZ* gene (Fig. 4c, d) could explain the significant increase in $N_2O$ emissions with N additions in the tropical forest soil (Han et al., 2018). In contrast, an opposite pattern with a higher *nosZ* gene abundance but a lower *nirK* gene abundance was found in Masson pine forest soil with low $N_2O$ emissions (Li et al., 2019a), suggesting that *nirK*- and *nosZ*-denitrifiers are critical in regulating $N_2O$ emission in different ecosystems. The decrease in the difference in *nosZ* gene abundance between the control and N-treated plots with time was possibly attributed to the tendency of microbial adaption to N addition. To some extent, the abundances of denitrifiers were related to their corresponding N transformation process.

### 4.3 Seasonal variations in N transformations under N additions

Seasonal patterns were more obvious for the N transformations in the second year of N additions. The $R_m$, $R_n$ and $R_l$ were apparently higher in the wet season than in the dry season (Fig. 2c, d and Fig. 3e), suggesting that soil temperature and moisture were the critical environmental factors affecting N transformations (Li et al., 2018a). Similar seasonal patterns have been documented in previous studies (Zhang et al., 2008b;Contosta et al., 2011;Li et al., 2014). In the dry season, the HN addition decreased the MBC and MBN by 15.1% and 14.5 respectively (Table S3), and the low temperature and precipitation suppressed the microbial biomass and activity and then depressed the N mineralization (Contosta et al.,



2011;Chen et al., 2017). Our results indicated that the lower soil C:N ratio with N enrichment was the dominant factor that increased N availability ($R_m$ and $R_n$), and subsequently led to higher N losses ($N_2O$ emission and nitrate leaching) in the dry season (Table 1a). In contrast, the factors controlling the processes of N transformation in the wet season were more complicated, including soil pH, inorganic N content, microbial biomass, SWC, TN, SOC and N-related functional gene abundance (Table 1b). Among them, the soil microbial biomass in the wet season was higher due to the higher SWC and temperature relative to that in the dry season in the forest soil (Deng et al., 2012), which could drive N mineralization and nitrification (Templer et al., 2005) and then form more large nitrate leaching.

## 4.4 The interactions between soil N transformations and abiotic/biotic conditions

In the dry season, the variations in $R_m$, $R_n$ and $R_l$ exhibited significant negative correlations with the soil C:N ratio (Table 1a), suggesting that the C:N ratio was a dominant factor determining soil N dynamics (Fang et al., 2011;Liu et al., 2017). In this study, the HN addition decreased the soil C:N ratio, which was consistent with the results of a previous study in an acidic forest soil (Shi et al., 2018). Significant positive correlations between the C:N ratios and the abundances of AOA, AOB, and *nos*Z-$N_2O$ reducers were observed (Table 1a), indicating that a low C:N ratio had a negative effect on N-related functional microbes. AOA abundance was positively correlated with soil $NO_3^-$-N concentration ($R = 0.47$, $P < 0.05$), which was in accordance with the previous studies (Hu et al., 2013;Tang et al., 2016). In addition, AOA abundance was negatively correlated with soil pH (Li et al., 2019a), indicating that AOA could adapt to the strong acidic tropical forest soil.

In the wet season, AOB *amoA* and *nos*Z gene abundances were positively related to soil $NH_4^+$-N contents and pH but negatively related to soil $NO_3^-$-N contents (Fig. 5 and Table 2b), indicating that the lower pH and accumulation of soil $NO_3^-$-N with N addition might result in decreases in the AOB *amoA* and *nos*Z gene abundances. There was no positive relationship between AOA abundance and $R_n$ in the study period (Table 2b), which disagreed with the results of a previous study showing that the abundance of AOA *amoA* was significantly correlated with gross nitrification rates in this acidic forest soil using the [15]N isotope dilution method (Isobe et al., 2012). This difference could be possibly explained by the net nitrification measurements underestimating gross nitrification (Li et al., 2018b). Furthermore, $N_2O$ emissions had significantly negative correlation with *nos*Z gene abundance ($R = -0.47$, $P < 0.05$) in the wet season (Table 2b). It was reported that decreasing *nos*Z gene abundance with N addition was the major factor resulting in low $N_2O$ consumption and high $N_2O$ emissions in the acidic forest soil (Zhang et al., 2008a). In addition, the findings also indicated that $R_m$, $R_n$ and $R_l$ were significantly and negatively correlated with soil $NH_4^+$-N content (Fig. 5), and the possible explanation was that $NH_4^+$-N could have a negative feedback on soil N mineralization (Geisseler et al., 2010;Zhang et al., 2018b). Interestingly, $R_m$, $R_n$ and $R_l$ were significantly and negatively correlated with soil pH, which contrasted with the results of previous studies (Fu et al., 1987;Kemmitt et al., 2006). The most reasonable explanation is that soil pH has a negative correlation with soil N transformation in strongly acidic soils (pH < 4.0), which is likely

due to the highest nitrification rates existing in the soils with lower pH (Booth et al., 2005).

The PLS-PM showed that the inorganic N had significantly negative direct effects on the N-related functional gene abundance (Fig. 6), suggesting that the functional microorganisms were more sensitive to N addition, and ongoing N deposition had significant negative effects on soil functional microbes

(Zhang et al., 2018c). These negative effects did not cause significant negative effects on N transformations in the study period, which was possibly explained by the microbial function redundancy or buffer capacity of the acidic forest soil. In addition, we found that the microbial biomass was the dominant factor driving N transformations in the wet season (Fig. 6), suggesting that microbes played a critical role in driving the processes of N transformation (Li et al., 2019b). However, it was previously

found that a 13-year N addition significantly decreased the MBC and MBN in adjacent forest soil (Wang et al., 2018b), implying that chronic N deposition would have a negative effect on soil N transformations.

## 5 Conclusions

The addition of N increased the *in situ* net mineralization, nitrification, inorganic N leaching rate, and $N_2O$ emission during the short term, which supported the traditional N saturation hypothesis. To some extent, the alterations of functional gene abundance with N additions were related to the corresponding

processes of N transformation. The variations in $R_m$, $R_n$ and $R_l$ exhibited different seasonal patterns. They were higher in the wet season than in the dry season. The C:N ratio was the dominant driving factor of N transformations in the dry season, while the biotic factors (microbial biomass) played an important role in accelerating N transformations in the wet season. According to the PLS-PM analysis, N additions had negative effects on the abundance of N-related functional genes in the dry season, which implies that

chronic N deposition poses a potential risk to forest ecosystem functions.

*Data availability.* All the relevant data are presented in the paper and supplementary materials.

*Author contributions.* WS designed the study, planned the field experiments and obtained research

funding. YN carried out the experiment and analyzed the data. YN, WS and MW wrote the manuscript. XH provided the $N_2O$ observations and guidance on their interpretation. JC helped in the field experiments of N transformation (*in situ* $R_m$, $R_n$ and $R_l$) and provided part of the data. All the authors provided feedback and gave constructive suggestions on the manuscript.

*Competing interests.* The authors declare that they have no conflict of interest.

*Acknowledgements.* We would like to thank for Dr. Wei Zhang for her assistance of soil sampling and



data collected. We thank Dr. Suping Liu and Ms. Zhuang Ni, Wenjuan Wang, Shaoyun Lv for their helpful assistance in the laboratory analysis. This study was financially supported by the National Natural Science Foundation of China (31600382, 31425005 and 31290222), the National Ten Thousand Talents Program, the Guangdong Province Baiqianwan Talents Program, the open project of Guangdong Provincial Key Laboratory of Plant Resources (2017B030314023), and the National Key Research and Development Program of China (2017YFD0202100).

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





Table 1 The linear correlation $R$ and $P$ value among soil properties, N-related functional genes and N transformation processes in the dry (a) and wet (b) seasons. The gray region represents the $R$ value, and the white region represents the $P$ value.

**(a)**

| Dry season | C/N | PH | NH$_4^+$-N | NO$_3^-$-N | MBC | MBN | TN | TP | SWC | SOC | AOA | AOB | nirK | nosZ | R$_m$ | R$_n$ | N$_2$O | R$_l$ |
|---|---|---|---|---|---|---|---|---|---|---|---|---|---|---|---|---|---|---|
| C/N | | 0.61 | 0.03 | 0.36 | 0.91 | 0.34 | 0.02 | 0.14 | 0.06 | 0.00 | 0.03 | 0.00 | 0.30 | 0.02 | 0.03 | 0.03 | 0.57 | 0.42 |
| PH | -0.11 | | 0.83 | 0.02 | 0.37 | 0.72 | 0.02 | 0.97 | 0.18 | 0.11 | 0.00 | 0.58 | 0.41 | 0.71 | 0.94 | 0.94 | 0.83 | 0.44 |
| NH$_4^+$-N | -0.44 | -0.05 | | 0.50 | 0.47 | 0.15 | 0.66 | 0.23 | 0.20 | 0.19 | 0.40 | 0.23 | 0.76 | 0.19 | 0.22 | 0.22 | 0.21 | 0.55 |
| NO$_3^-$-N | -0.20 | -0.47 | 0.15 | | 0.10 | 0.17 | 0.19 | 0.60 | 0.20 | 0.76 | 0.02 | 0.30 | 0.52 | 0.27 | 0.22 | 0.22 | 0.38 | 0.17 |
| MBC | -0.03 | -0.19 | -0.16 | 0.34 | | 0.00 | 0.07 | 0.86 | 0.21 | 0.25 | 0.26 | 0.36 | 0.08 | 0.27 | 0.56 | 0.56 | 0.98 | 0.74 |
| MBN | -0.20 | -0.08 | -0.31 | 0.29 | 0.69 | | 0.92 | 0.32 | 0.07 | 0.64 | 0.94 | 0.89 | 0.20 | 0.57 | 0.77 | 0.77 | 0.62 | 0.88 |
| TN | 0.47 | -0.48 | -0.09 | 0.27 | 0.37 | 0.02 | | 0.02 | 0.60 | 0.00 | 0.00 | 0.01 | 0.15 | 0.02 | 0.28 | 0.28 | 0.79 | 0.94 |
| TP | 0.31 | -0.01 | -0.25 | -0.11 | -0.04 | -0.21 | 0.49 | | 1.00 | 0.02 | 0.43 | 0.26 | 0.75 | 0.41 | 0.15 | 0.15 | 0.68 | 0.60 |
| SWC | -0.39 | -0.29 | -0.27 | 0.27 | 0.26 | 0.38 | -0.11 | 0.00 | | 0.16 | 0.99 | 0.09 | 0.27 | 0.12 | 0.33 | 0.33 | 0.12 | 0.39 |
| SOC | 0.83 | -0.34 | -0.28 | 0.06 | 0.24 | -0.10 | 0.88 | 0.46 | -0.30 | | 0.00 | 0.00 | 0.13 | 0.00 | 0.07 | 0.07 | 0.61 | 0.61 |
| AOA | 0.44 | -0.64 | -0.18 | 0.47 | 0.24 | 0.02 | 0.56 | 0.17 | 0.00 | 0.59 | | 0.01 | 0.29 | 0.04 | 0.61 | 0.61 | 0.97 | 0.34 |
| AOB | 0.58 | -0.12 | -0.25 | -0.22 | 0.20 | 0.03 | 0.51 | 0.24 | -0.36 | 0.65 | 0.49 | | 0.01 | 0.00 | 0.18 | 0.18 | 0.71 | 0.68 |
| nirK | 0.22 | -0.18 | -0.07 | -0.14 | 0.36 | 0.27 | 0.30 | -0.07 | -0.23 | 0.32 | 0.23 | 0.51 | | 0.00 | 0.50 | 0.50 | 0.57 | 0.36 |
| nosZ | 0.48 | -0.08 | -0.28 | -0.24 | 0.23 | 0.12 | 0.47 | 0.17 | -0.33 | 0.57 | 0.43 | 0.89 | 0.67 | | 0.38 | 0.38 | 0.57 | 0.97 |
| R$_m$ | -0.45 | -0.02 | 0.26 | 0.26 | 0.13 | 0.06 | -0.23 | -0.30 | 0.21 | -0.38 | -0.11 | -0.28 | -0.14 | -0.19 | | 0.00 | 0.84 | 0.00 |
| R$_n$ | -0.45 | -0.02 | 0.26 | 0.26 | 0.13 | 0.06 | -0.23 | -0.30 | 0.21 | -0.38 | -0.11 | -0.28 | -0.14 | -0.19 | 1.00 | | 0.84 | 0.00 |
| N$_2$O | -0.12 | -0.05 | -0.27 | -0.19 | 0.01 | 0.11 | -0.06 | -0.09 | 0.32 | -0.11 | 0.01 | 0.08 | -0.12 | 0.12 | -0.04 | -0.04 | | 0.82 |
| R$_l$ | -0.17 | -0.16 | 0.13 | 0.29 | 0.07 | 0.03 | -0.02 | -0.11 | 0.18 | -0.11 | 0.20 | -0.09 | -0.19 | -0.01 | 0.85 | 0.85 | 0.05 | |




**(b)**

| Wet season | C/N | PH | NH$_4^+$-N | NO$_3^-$-N | MBC | MBN | TN | TP | SWC | SOC | AOA | AOB | *nirK* | *nosZ* | R$_m$ | R$_n$ | N$_2$O | R$_l$ |
|---|---|---|---|---|---|---|---|---|---|---|---|---|---|---|---|---|---|---|
| C/N | | 0.63 | 0.56 | 0.47 | 0.15 | 0.64 | 0.69 | 0.88 | 0.86 | 0.00 | 0.67 | 0.18 | 0.17 | 0.18 | 0.63 | 0.63 | 0.05 | 0.45 |
| PH | 0.10 | | 0.00 | 0.00 | 0.19 | 0.00 | 0.01 | 0.48 | 0.00 | 0.18 | 0.32 | 0.00 | 0.32 | 0.00 | 0.00 | 0.00 | 0.15 | 0.00 |
| NH$_4^+$-N | -0.13 | 0.69 | | 0.00 | 0.04 | 0.00 | 0.06 | 0.81 | 0.00 | 0.07 | 0.25 | 0.01 | 0.79 | 0.00 | 0.00 | 0.00 | 0.00 | 0.00 |
| NO$_3^-$-N | -0.15 | -0.78 | -0.76 | | 0.24 | 0.00 | 0.00 | 0.62 | 0.00 | 0.09 | 0.15 | 0.01 | 0.77 | 0.00 | 0.00 | 0.00 | 0.07 | 0.00 |
| MBC | 0.30 | -0.27 | -0.43 | 0.25 | | 0.00 | 0.05 | 0.48 | 0.00 | 0.01 | 0.68 | 0.77 | 0.41 | 0.09 | 0.01 | 0.01 | 0.01 | 0.01 |
| MBN | 0.10 | -0.65 | -0.71 | 0.77 | 0.61 | | 0.00 | 0.10 | 0.00 | 0.00 | 0.21 | 0.25 | 0.99 | 0.00 | 0.00 | 0.00 | 0.05 | 0.00 |
| TN | -0.09 | -0.51 | -0.39 | 0.64 | 0.40 | 0.71 | | 0.12 | 0.00 | 0.00 | 0.38 | 0.41 | 0.95 | 0.13 | 0.01 | 0.01 | 0.07 | 0.03 |
| TP | -0.03 | -0.15 | -0.05 | 0.11 | 0.15 | 0.35 | 0.32 | | 0.26 | 0.30 | 0.28 | 0.54 | 0.99 | 0.29 | 0.34 | 0.34 | 0.92 | 0.24 |
| SWC | 0.04 | -0.65 | -0.69 | 0.74 | 0.59 | 0.91 | 0.77 | 0.24 | | 0.00 | 0.09 | 0.37 | 1.00 | 0.00 | 0.00 | 0.00 | 0.01 | 0.00 |
| SOC | 0.68 | -0.28 | -0.38 | 0.36 | 0.53 | 0.60 | 0.66 | 0.22 | 0.59 | | 0.82 | 0.63 | 0.35 | 0.02 | 0.03 | 0.03 | 0.00 | 0.03 |
| AOA | 0.09 | 0.21 | 0.24 | -0.30 | 0.09 | -0.27 | -0.19 | -0.23 | -0.36 | -0.05 | | 0.24 | 0.75 | 0.77 | 0.26 | 0.26 | 0.85 | 0.30 |
| AOB | 0.28 | 0.59 | 0.50 | -0.49 | -0.06 | -0.24 | -0.17 | -0.13 | -0.19 | 0.10 | 0.25 | | 0.34 | 0.34 | 0.02 | 0.02 | 0.87 | 0.02 |
| *nirK* | -0.29 | -0.21 | -0.06 | 0.06 | 0.18 | 0.00 | 0.01 | 0.00 | 0.00 | -0.20 | -0.07 | -0.20 | | 0.51 | 0.38 | 0.38 | 0.46 | 0.34 |
| *nosZ* | -0.29 | 0.64 | 0.63 | -0.58 | -0.35 | -0.66 | -0.32 | -0.22 | -0.57 | -0.47 | -0.06 | 0.20 | 0.14 | | 0.00 | 0.00 | 0.02 | 0.00 |
| R$_m$ | 0.10 | -0.56 | -0.64 | 0.68 | 0.50 | 0.71 | 0.50 | 0.20 | 0.59 | 0.45 | -0.24 | -0.47 | 0.19 | -0.59 | | 0.00 | 0.04 | 0.00 |
| R$_n$ | 0.10 | -0.56 | -0.64 | 0.68 | 0.50 | 0.71 | 0.50 | 0.20 | 0.59 | 0.45 | -0.24 | -0.47 | 0.19 | -0.59 | 1.00 | | 0.04 | 0.00 |
| N$_2$O | 0.40 | -0.31 | -0.55 | 0.38 | 0.51 | 0.41 | 0.37 | 0.02 | 0.53 | 0.56 | 0.04 | -0.04 | -0.16 | -0.47 | 0.43 | 0.43 | | 0.03 |
| R$_l$ | 0.16 | -0.56 | -0.64 | 0.63 | 0.49 | 0.67 | 0.45 | 0.25 | 0.57 | 0.45 | -0.22 | -0.49 | 0.20 | -0.58 | 0.98 | 0.98 | 0.44 | |





## Figure Legends

Fig. 1. Changes in soil $NH_4^+$-N (a) and $NO_3^-$-N (b) contents in the soils at different samplings. Bars represent standard errors of the mean (n=3). Significance levels are indicated by $*P < 0.05$, $**P < 0.01$.

Fig. 2. The variation in the *in situ* mineralization rate ($R_m$) (a) and nitrification rate ($R_n$) (b) in the soils at different samplings. The $R_m$ (c) and $R_n$ (d) in the dry and wet seasons. Bars represent standard errors of the mean (n=3). Significance levels are indicated by $*P < 0.05$, $**P < 0.01$.

Fig. 3. Dynamics of the *in situ* $NH_4^+$-N(a), $NO_3^-$-N(b) and total inorganic N leaching rates ($R_l$) (c) in the soils at different samplings. d. The correlation between the rates of nitrate leaching and inorganic nitrogen leaching ($R_l$). e. The variation in $R_l$ in the dry and wet season. f. The rates of annual $N_2O$ emission with N addition. Bars represent standard errors of the mean (n=3). Significance levels are indicated by $*P < 0.05$, $**P < 0.01$.

Fig. 4. Responses of functional genes (AOA *amoA*(a), AOB *amoA*(b), *nirK*(c), and *nosZ*(d)) to N deposition. Bars represent standard errors of the mean (n=3). Significance levels are indicated by $*P < 0.05$, $**P < 0.01$.

Fig. 5 Redundancy analysis (RDA) among environmental variables ($NH_4^+$-N, $NO_3^-$-N, pH, SOC, TN, C/N and SWC), functional genes (AOA *amoA*, AOB *amoA*, *nirK*, and *nosZ*) and soil N transformation rates ($R_m$, $R_n$, $R_l$ and $N_2O$ emission) in the wet season. Values on the axes indicated the percentages of total variation explained by each axis.

Fig. 6. Directed graph of the partial least squares path model (PLS-PM) of the inorganic N ($NH_4^+$-N and $NO_3^-$-N), soil conditions (pH, SOC, TN, C/N and SWC), microbial biomass (MBC and MBN), and the abundances of functional genes (AOA *amoA*, AOB *amoA*, *nirK*, and *nosZ*) effects on soil N transformation rates ($R_m$, $R_n$, $R_l$ and $N_2O$ emission) in the wet season. Path coefficients and explained variability ($R^2$) reflecting in the width of the arrow were calculated after 1000 bootstraps. The blue and red representing positive and negative effects, respectively. Solid arrows indicated $P < 0.05$; and dashed arrows indicate $P > 0.05$. The model was assessed using the Goodness of Fit (GoF). $* P < 0.05$, $**P < 0.01$, $***P < 0.001$.



Fig. 1

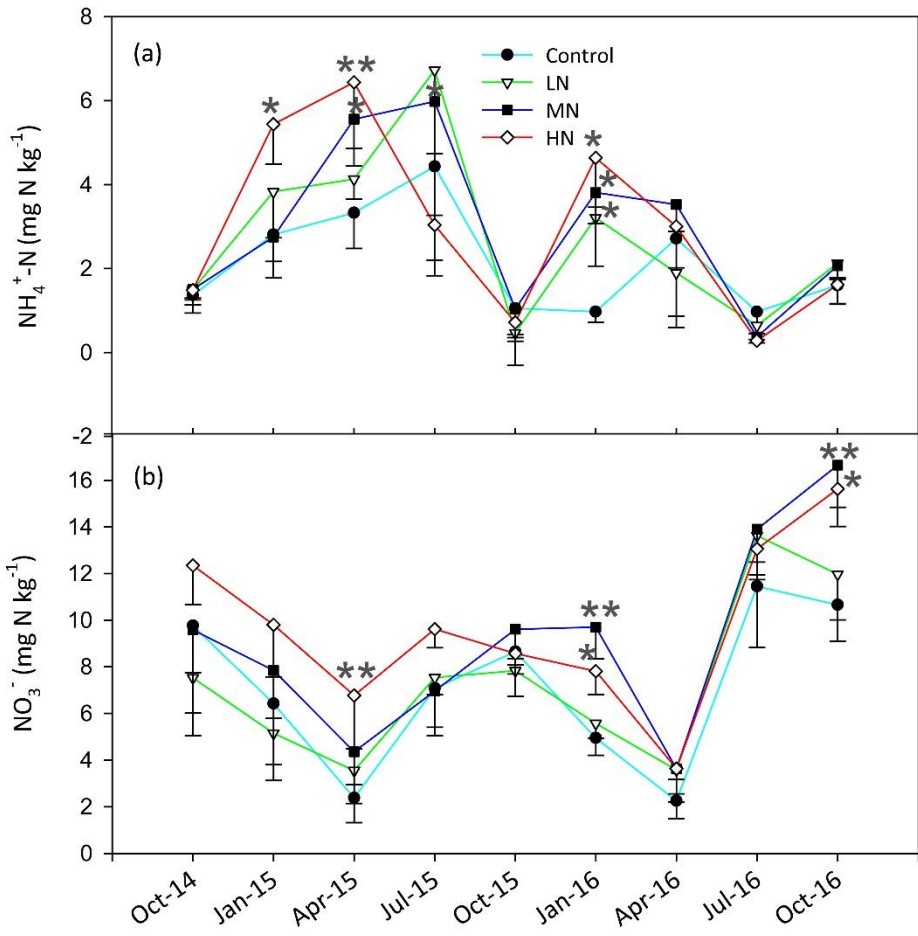



Fig. 2

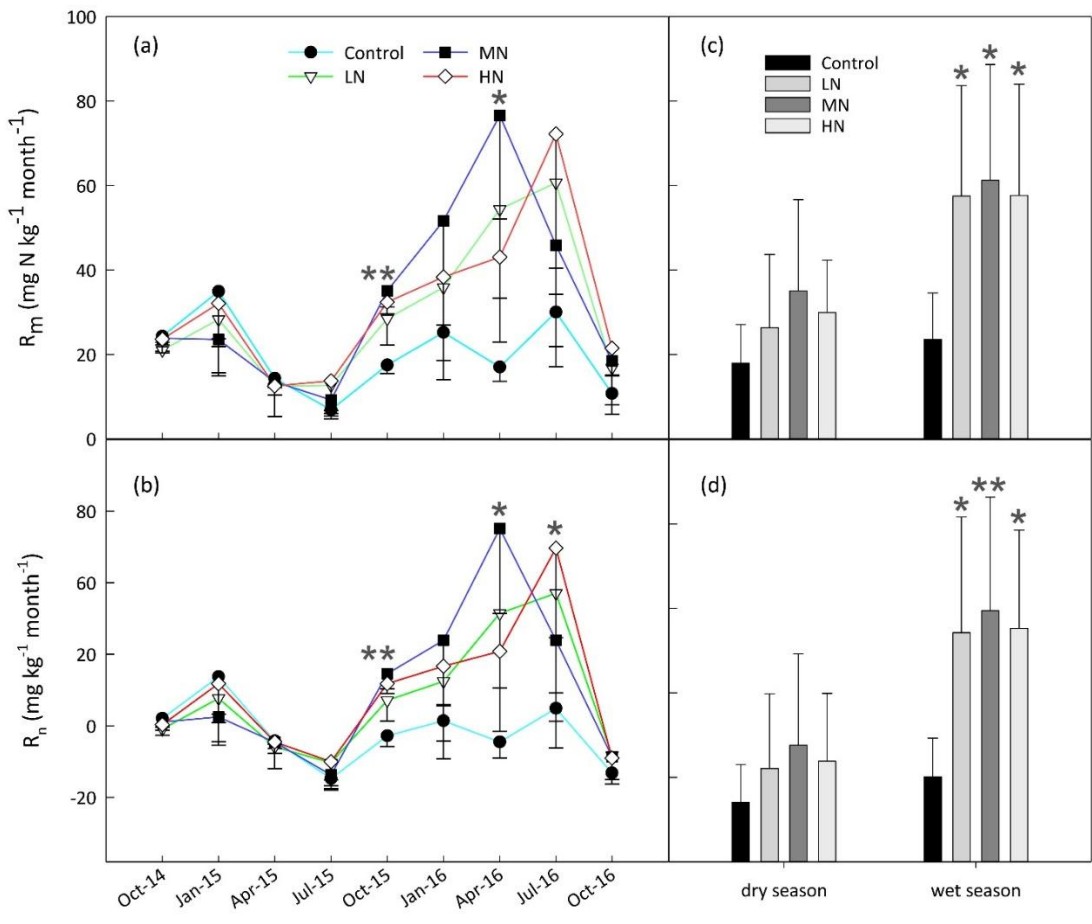



Fig. 3

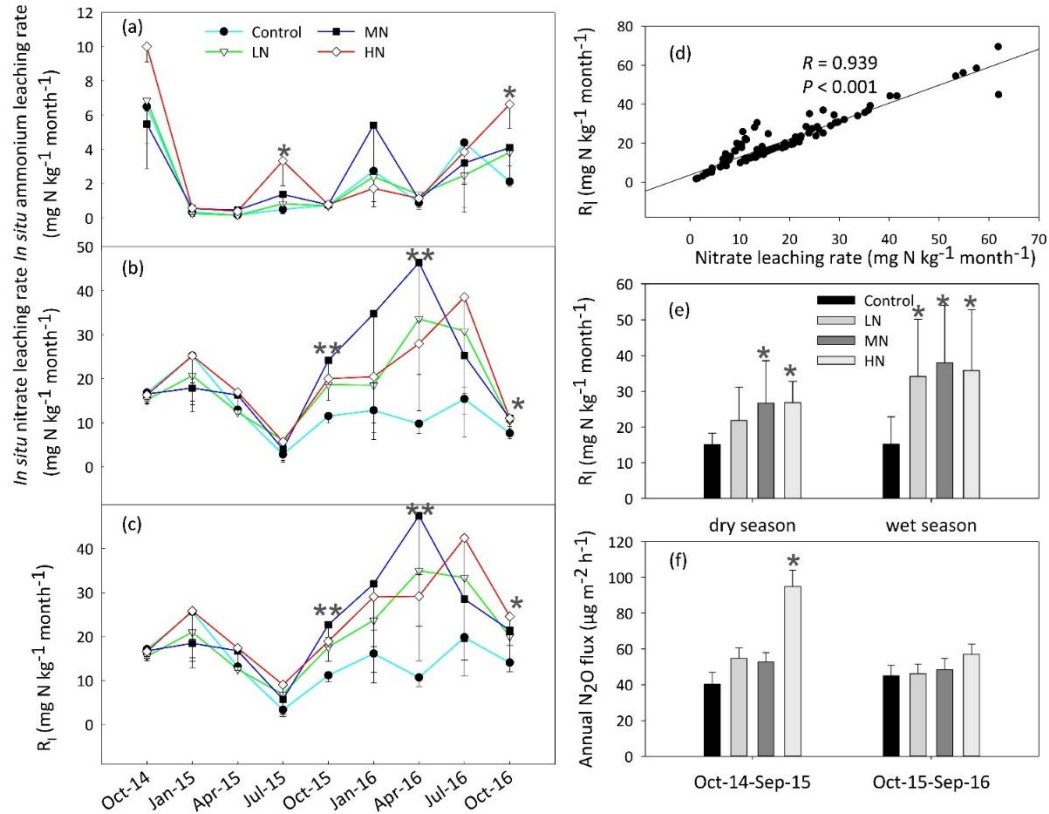



Fig. 4

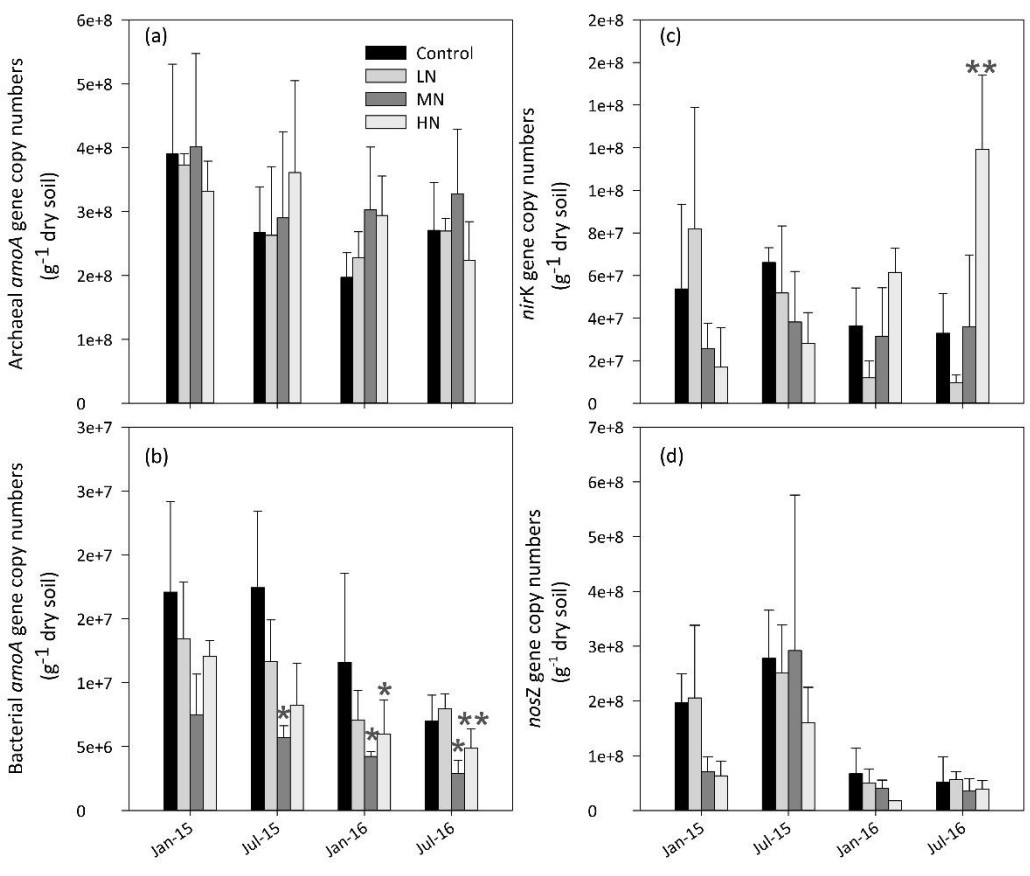



Fig. 5

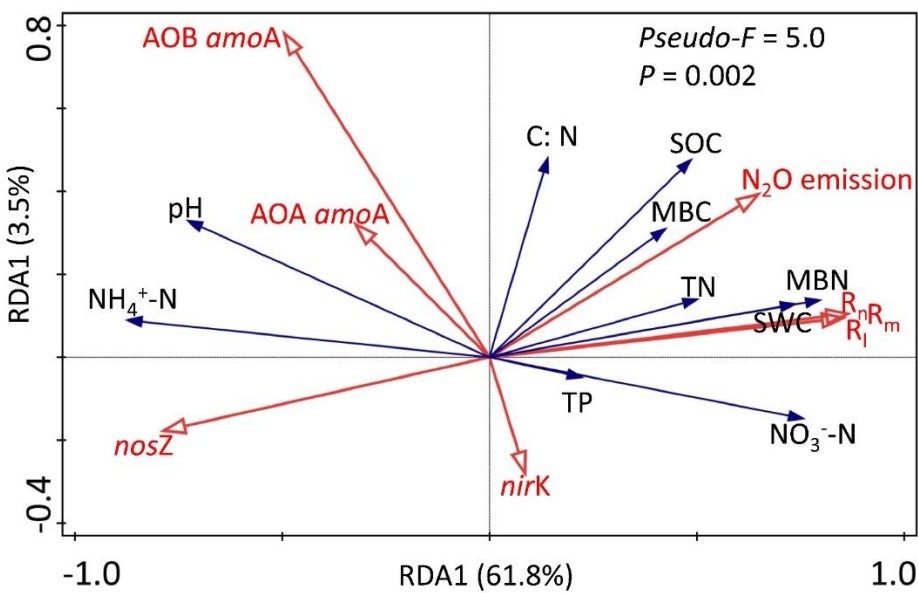





Fig. 6

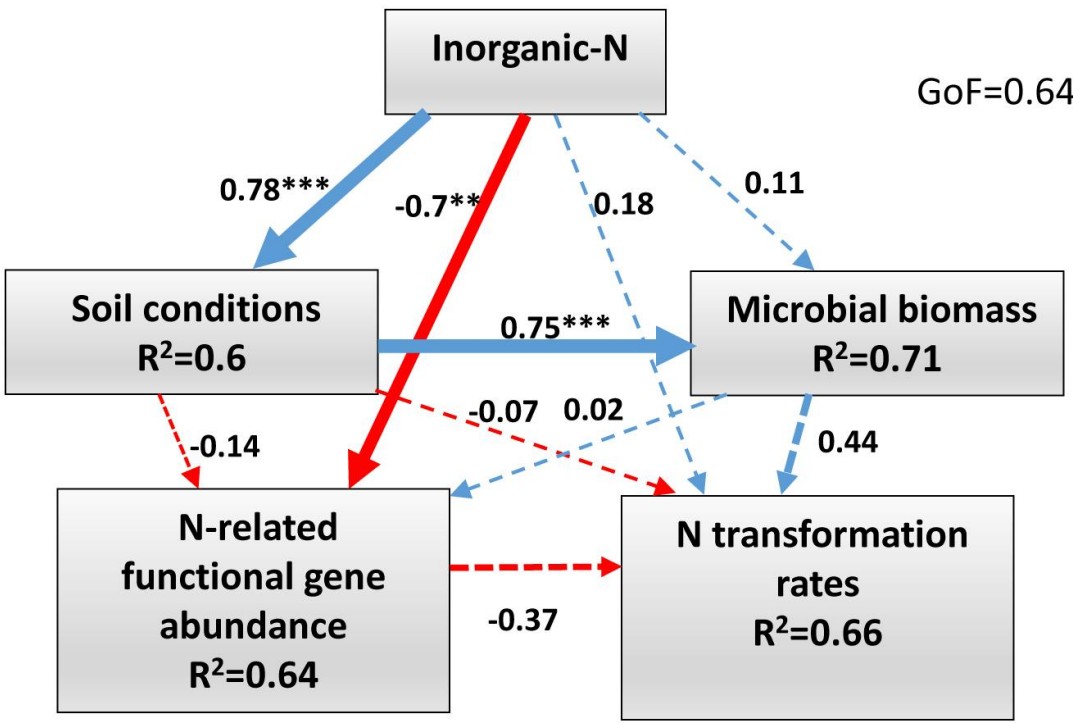