# Peer review of "The simulated N deposition accelerates net N mineralization and nitrification in a tropical forest soil"

_Biogeosciences, 2019_

## Referee Comment (RC1) · Anonymous Referee #1 · 26 Apr 2019

This paper examined the effect of N addition on soil N transformation processes. The information is valuable to our understanding of how increasing N deposition could change soil microbes and the N process they drive. However, I believe the paper should be significantly revised before publication. First, the difference between net N process and gross N process should be carefully discussed. Second, the discussion section is still the re-statement of the findings, but the underlying mechanisms of the findings were not analyzed enough. Especially, if there are inconsistent results currently in different studies, it is better to explain why the difference was observed. If this study only presents the difference, it only increases the uncertainty of current findings, but could not contribute to improve our understanding of the current findings. Please

see the specific comments:

P2L9-10: This sentence is not clear. Rm and Rn could be driven by soil microbes, but what do you mean by saying they were driven by higher microbial biomass? you mean positive relationship between Rm and MBC? P2L13: Rm and Rn are only net N transformation rates, I don't think they are equal to soil N availability. Inorganic N content is a better proxy of soil N availability. P2L18: significantly P2L20: what do you mean by saying a rate is "delayed"? P3L15-30: this paragraph listed out some papers with different results. However, it is better to summarize these results and analyze why these results were different. For example, there are more similar studies available, why did the authors choose to mention these single papers? Did they all examined net N transformation rate? P4L15-24: Yes, gross N transformation rate is controlled by environmental factors and microbial properties. However, net N transformation rate is the results of changes in both input and output. If the gross N production rate is increased, or the N consumption rate is reduces, both could cause the increase of a net N rate. Therefore, it is better to differentiate gross N transformation and net N transformation in the introduction section. P4L25-L30: Again, N availability is about N pool size, while N transformation is about N dynamic. They are not the same thing. P6L15: More information on N2O emission is needed. How many times were N2O emission measured in each month? How was annual rate calculated? P6L27: was a dividing factor used to calculate MBC and MBN? P6: It should be clarified that N inorganic N was added into the PVC tubes when N addition treatment was carried out. P7L8: repeated measures ANOVA should be used. P7L14: Is the premise of the PLS-PM method satisfied? P10L5: Again, the difference between gross rate and net rate should be discussed. The promotion of net N mineralization was due to the promotion of gross N mineralization? or due to the reduction in immobilization? or other Loss fluxes? P11L10: NH3 should be NH4+? P11L14: I wonder if the authors could dig more on the reasons of different findings rather than just saying the difference was due to different systems. If this study cannot contribute to our understanding of the reasons of current different findings, this study can only increase the uncertainty of our

understand on N cycling. P11L23-24: what is the climate of the Masson forest in Li et al. 2019? Because nosZ is mainly affected by soil moisture conditions, it is important to know the climate information. P12L1-5: The authors only stated the results again, but did not discuss why the controlling factors were different between the two seasons. I imagine soil C:N would not change seasonally due to the large pool size. Why wasn't it a controlling factor in the wet season? P12L20-35: I think some of the discussions here should be mentioned earlier and this section should be re-organized to be more logical. For example, the difference between gross and net rate; the effect of NosZ on N2O, the effect of moisture (wet season) on NosZ and N2O. These information are important factors for understanding the underlying mechanisms of the findings and should be carefully analyzed when the findings were discussed. P13L5-10: For net N mineralization, it could be possible that both gross N mineralization and immobilization were suppressed, while net N mineralization did not change much. Then it does not mean the negative effects of N addition on soil microbes did not affect N transformation.

---

## Referee Comment (RC2) · Anonymous Referee #2 · 10 Jun 2019

Due to the complexity of nitrogen cycling in terrestrial ecosystems, it deserves to explore how elevated nitrogen deposition affects soil N transformations in the N-rich soil of tropical forests. Overall, this manuscript was well written and easy to read, but the current version is suffering from some critical defects. First, this study measured the net mineralization and nitrification, completely different from gross mineralization and nitrification. To this point, the title of this study is not appropriate, because net mineralization and nitrification actually include the balance of various transformation processes such as ammoniation and immobilization, which conceals real nitrogen transformation processes. Second, the descriptions in Methods are not detailed and thus affect understanding of the results. For example, the descriptions about the specific time for nitrocreate

gen addition and sampling soil cores for net mineralization were unclear. Considering net mineralization is the difference of ammonium concentrations between 30 days, the time for nitrogen addition and the sampling of two soil cores is very important. If the sampling of second soil cores was just after nitrogen addition, mineralization could be overestimated because added N contributed to increase in soil ammonium concentrations. Third, it is well known that nitrogen addition will lead to soil acidification. However, this study did not separate from inorganic nitrogen input from its acidification (also see Fig. 6). This strongly reduces the importance of this study, e.g. both low pH and higher inorganic nitrogen concentrations can show negative effects on nitrogen transformations. Fourth, it is very good to include the measurements of N-related functional gene abundance, but it is a pity that N-related functional gene abundance was not related with the specific nitrogen transformation processes. As a result, it is difficult to make a microbial mechanism explanation for net mineralization and nitrification. Before the manuscript is accepted to publish, the above issues should be well clarified.

————————————————————

---

## Author Comment (AC1) · 30 Jun 2019

Yanxia Nie and co-authors
Correspondence to: Weijun Shen (shenweij@scbg.ac.cn)

***Response to Anonymous Referee #1***

**Reviewer comment:** *This paper examined the effect of N addition on soil N transformation processes. The information is valuable to our understanding of how increasing N deposition could change soil microbes and the N process they drive. However, I believe the paper should be significantly revised before publication.*

**Response:** Thank you for the positive evaluation of our work. The manuscript has been revised based on the suggestions of yours and another reviewer's. We hope that you would find the revision satisfactory.

**Reviewer comment:** *First, the difference between net N process and gross N process should be carefully discussed.*

**Response:** In this study, we actually only measured and reported the net N mineralization and nitrification rates estimated using the field incubation method, which are different from the gross N mineralization and nitrification rates often estimated using the $^{15}N$ dilution method as the reviewer pointed out. We carefully distinguished them in discussions.

**Reviewer comment:** *Second, the discussion section is still the re-statement of the findings, but the underlying mechanisms of the findings were not analyzed enough. Especially, if there are inconsistent results currently in different studies, it is better to explain why the difference was observed. If this study only presents the difference, it only increases the uncertainty of current findings, but could not contribute to improve our understanding of the current findings.*

**Response:** Thanks for the constructive suggestion. We have revised the discussion section by eliminating the repetition of results and providing more mechanistic explanations for the different results found between our study and previous studies. For example, in the last paragraph of page 14, we compared our results with those of a previous study and explained why different results were obtained between the two studies.

*Specific comments:*

*P2L9-10: This sentence is not clear. Rm and Rn could be driven by soil microbes, but what do you mean by saying they were driven by higher microbial biomass? You mean positive relationship between Rm and MBC?*

**Response:** Yes, we meant to say there were significantly positive relationships between net N processes ($R_m$ and $R_n$) and microbial biomass (MBC and MBN), but we used an inaccurate word as the reviewer noticed. We have corrected the sentence

as "The $R_m$ and $R_n$ were mainly associated with the N addition-induced lower C:N ratio in the dry season but with higher microbial biomass in the wet season".

*P2L13: Rm and Rn are only net N transformation rates, I don't think they are equal to soil N availability. Inorganic N content is a better proxy of soil N availability.*
**Response:** Yes, $R_m$ and $R_n$ are the net N mineralization and nitrification rates estimated as the difference of inorganic N contents measured at the beginning and the end of the *in situ* incubation divided by the duration of the incubation period. So these rates do not directly reflect soil N availability; they reflect changes in soil inorganic N content over a period of time (in our study 30 days). We have corrected the sentence as "N additions significantly facilitated $R_m$, $R_n$, $R_l$ and $N_2O$ emission".

*P2L18: significantly*
**Response:** Thanks! The correction has been made as suggested.

*P2L20: what do you mean by saying a rate is "delayed"?*
**Response:** In this study, we observed that the responses of soil net N transformation rates (i.e., *in situ* $R_m$ and $R_n$) to N additions were not significant in the first year but became so in the second year. We therefore meant to express that we observed a delayed response. But the description was not clear enough to readers. We have revised the sentence as "The responses of soil net N transformations (*in situ* $R_m$, and $R_n$) and $R_l$ to N additions were negligible during the first year of N inputs" in P2L6.

*P3L15-30: this paragraph listed out some papers with different results. However, it is better to summarize these results and analyze why these results were different. For example, there are more similar studies available, why did the authors choose to mention these single papers? Did they all examined net N transformation rate?*

**Response:** In this paragraph, we aimed to make two points: 1) relatively fewer studies addressing N addition effects on soil N transformations have been conducted in tropical forest ecosystems; 2) existing such studies have received inconsistent results for a variety of reasons. We listed 3 pairs of such N addition experiments that were conducted in the same region but differed in forest age, duration of N addition, and soil properties. We further clarified the descriptions and added a summarizing sentence to point out the main factors affecting soil N transformation responses to N additions in tropical forests. We also specified whether net or gross N transformation rates were measured in these studies.

*P4L15-24: Yes, gross N transformation rate is controlled by environmental factors and microbial properties. However, net N transformation rate is the results of changes in both input and output. If the gross N production rate is increased, or the N consumption rate is reduces, both could cause the increase of a net N rate. Therefore, it is better to differentiate gross N transformation and net N transformation in the introduction section.*
**Response:** Thank you very much for the suggestions. We clearly differentiated the

gross and net N transformation rates in the Introduction section during this revision. The net N transformation rates assessed in our field incubation study are actually different from the gross N transformation rates assessed with the $^{15}$N dilution method in lab incubations. Please also see our response to your second comment for further clarifications between the *in situ* net and the lab gross N transformation rates. .

*P4L25-L30: Again, N availability is about N pool size, while N transformation is about N dynamic. They are not the same thing.*
**Response:** Agreed. We have corrected it as "net N transformation processes (i.e., N mineralization and nitrification) and nitrate leaching and $N_2O$ emission".

*P6L15: More information on N2O emission is needed. How many times were N2O emission measured in each month? How was annual rate calculated?*
**Response:** Following your suggestion, more information on $N_2O$ emission measurements and calculations have been added during this revision (P7L23-30). Briefly, soil $N_2O$ emissions were monitored using the closed chamber method. The gas samples were taken twice each month from October 2014 to September 2016, and $N_2O$ concentrations were analyzed with a gas chromatograph. The rate of $N_2O$ emission was calculated using the following equation:

$$F = \rho \times \frac{V}{A} \times \frac{P}{P_0} \times \frac{T_0}{T} \times \frac{dC_1}{dt} \tag{4}$$

where F represents the $N_2O$ flux ($\mu g\ N\ m^{-2}\ h^{-1}$); $\rho$ the density of $N_2O$ under standard conditions ($mg\ L^{-1}$),, $V$ gas volume in the chamber ($m^3$), $A$ chamber coverage area ($m^2$), $P$ atmosphere pressure at the sampling time (Pa), $P_0$ standard atmosphere pressure (Pa), $T$ absolute temperature (K) at the sampling time, absolute temperature (K) under standard conditions, and $dC_1/dt$ the liner slope of gas concentration changes within the sampling time period. The annual rates of $N_2O$ emission ($kg\ N\ ha^{-1}\ yr^{-1}$) after N addition were calculated by linear interpolation between sampling dates in the two observation years: October 2014 to September 2015 and October 2015 to September 2016.

*P6L27: was a dividing factor used to calculate MBC and MBN?*
**Response:** Yes, we used 0.45 and 0.54 as the conversion factors for MBC and MBN, respectively. We have added the information and relevant citations (Brookes et al. 1985, Soil Biology and Biochemistry, 17: 837-842; Joergensen et al. 2011, Soil Biology and Biochemistry, 43: 873-876) in this revision.

*P6: It should be clarified that N inorganic N was added into the PVC tubes when N addition treatment was carried out.*
**Response:** This is a field N addition experiment. N solutions were sprayed evenly on the soil surface in plots of 225 $m^2$, as well as into the incubation PVC tubes installed in the 0-10 cm soil layer. We have added more specific information on the time of N additions and the field incubation occasions. Briefly, N additions were conducted at

the end of each month (around 24[th]) starting in September 2014. The field incubations for assessing net N transformation rates were conducted 9 times from September 2014 through October 2016, i.e., September 2014, December 2014, March 2015, June 2015, September 2015, December 2015, March 2016, June 2016, and September 2016. Each incubation was started a couple of days before the N addition date and lasted for 30 days.

*P7L8: repeated measures ANOVA should be used.*
**Response:** Agreed. Two-way repeated measures ANOVA was performed to examine the effects of N additions on soil $NH_4^+$-N and $NO_3^-$-N contents, N transformation rates and functional genes abundance over time. The results (Table S4 and S5) were listed in the section of supplementary material. We also added the corresponding description in the sections of Results and Discussion.

*P7L14: Is the premise of the PLS-PM method satisfied?*
**Response:** The Partial Least Squares Path Modeling (PLS-PM) method, a particularly useful statistical method for illuminating cause and effect relationships among observed and latent variables (Tenenhaus et al. 2005, Computational statistics and data analysis, 48: 159-205), is often used to explore the effects of environmental variables on soil microbial communities and N transformation process and further evaluate potential causal relationships between the variables (Fan et al. 2019, Soil Biology and Biochemistry, 130: 82-93; Dai et al. 2019, Geoderma, 337: 1116-1125). The premise of using this method is the small sample size (n < 200), and we are assuming that all latent variables are relevant. The model is assessed using the Goodness of Fit (GoF) statistic, and the value of goodness of fit in this study which is acceptable (Sanchez. 2013, PLS Path Modeling with R, 1-222).

*P10L5: Again, the difference between gross rate and net rate should be discussed. The promotion of net N mineralization was due to the promotion of gross N mineralization? or due to the reduction in immobilization? or other Loss fluxes?*
**Response:** As we responded before, the net N transformation rates measured in this study are different from the gross N transformation rates in terms of both quantity and the methodology used: net rates in our study are assessed from field incubation whereas gross rates are often assessed with the [15]N dilution method in lab incubation. Based on a recent [15]N dilution lab incubation study using the soil samples collected from the same experimental plots as in this study, Han et al. (2018) found that the gross N mineralization rate was stimulated whereas the gross N immobilization rate was suppressed by the N additions (Science of the Total Environment, 626: 1175-1187). Therefore, the stimulation of the net N mineralization rate observed in the second year of this study might be caused by both the increased gross N mineralization rate and the reduced gross immobilization rate. However, to establish a more rigorous link between the *in situ* net rates and lab incubation gross rates, a lab incubation on net rates as done in Lovett et al. (2004, Biogeochemistry, 67: 289-308) should be helpful. We have added these descriptions at the end of the first discussion

paragraph.

*P11L10: NH3 should be NH4+?*
**Response:** No! Here, it was ammonia ($NH_3$) rather than ammonium ($NH_4^+$). $NH_3$ was the direct substrate to ammonia monooxygenase (Suzuki, 1974, journal of bacteriology, 120: 556-558). However, in the acidic soils, $NH_3$ substrate availability significantly below the demand of AOB, but AOA had higher substrate affinity (Stopnišek et al., 2010, Applied and Environmental Microbiology, 76: 7626,).

*P11L14: I wonder if the authors could dig more on the reasons of different findings rather than just saying the difference was due to different systems. If this study cannot contribute to our understanding of the reasons of current different findings, this study can only increase the uncertainty of our understand on N cycling.*
**Response:** In this paragraph, we mainly wanted to explain why N additions had a positive (or neutral) impact on AOA abundance but a negative impact on AOB abundance by comparing with the results from a long-term (6-year) N addition experiment in a similar forest. Both decreased pH and $NH_4^+$ availability were the potential major contributors. We revised the whole paragraph by eliminating some repetitions of our results and the case study that is less comparable to our system. For example, the case study from the temperate steppe ecosystem (Zhang et al., 2018a, Applied and Soil Ecology, 130: 241-250), since it is a semiarid ecosystem with the dominant limiting factor being water availability.

*P11L23-24: what is the climate of the Masson forest in Li et al. 2019? Because nosZ is mainly affected by soil moisture conditions, it is important to know the climate information.*
**Response:** Thanks for your suggestion! The Masson forest site has a mean annual temperature of 17°C and mean annual precipitation of 1200–1400 mm. We have provided this information in the revision.

*P12L1-5: The authors only stated the results again, but did not discuss why the controlling factors were different between the two seasons. I imagine soil C:N would not change seasonally due to the large pool size. Why wasn't it a controlling factor in the wet season?*
**Response:** As the reviewer assumed, soil C:N ratio indeed did not exhibit a seasonal variation (Table. S3), but it was decreased by the N additions, and the decreased C:N ratio was the dominant cause for the increased Rm and Rn in the dry season. According to the RDA analysis, soil C:N ratio was also a controlling factor in the wet season. However, the importance of the C:N ratio in affecting N transformation rates was lower than the other factors such as microbial biomass and soil water content (Fig. 5). We therefore did not consider it as a dominant influential factor in the wet season.

*P12L20-35: I think some of the discussions here should be mentioned earlier and this section should be re-organized to be more logical. For example, the difference between gross and net rate; the effect of NosZ on N2O, the effect of moisture (wet*

*season) on NosZ and N2O. These information are important factors for understanding the underlying mechanisms of the findings and should be carefully analyzed when the findings were discussed. The whole paragraph need to be reorganized according to this suggestion.*

**Response:** Following your suggestion, we have discussed the difference between gross and net rates, the effect of *nosZ* gene abundance on $N_2O$ emission, and the effect of moisture (wet season) on *nosZ* gene abundance and $N_2O$ emission in the first and fifth paragraph of the Discussion section. We further clarified the discussions on the comparison between our findings and those of Isobe et al. (2012, FEMS Microbiology Ecology, 80: 193-203). Hopefully these revisions would make the section reads more logical.

*P13L5-10: For net N mineralization, it could be possible that both gross N mineralization and immobilization were suppressed, while net N mineralization did not change much. Then it does not mean the negative effects of N addition on soil microbes did not affect N transformation.*

**Response:** We specified that the N mineralization and nitrification rates we were referring to was the net N transformation rates, which were measured using the field incubation method in this study. We agree with the reviewer that N addition can alter both gross N mineralization rate and gross immobilization rate without altering the net rate, since net rate (e.g., net N mineralization rate) conceptually is the balance between gross mineralization rate and gross immobilization rate. Therefore, unchanged net rates do not mean unchanged gross rates, which were possibly suppressed because of the negative effects of N additions on soil microbes as the reviewer pointed out.

---

## Author Comment (AC2) · 30 Jun 2019

Yanxia Nie and co-authors
Correspondence to: Weijun Shen (shenweij@scbg.ac.cn)

***Response to Anonymous Referee #2***

***Reviewer comment:*** Due to the complexity of nitrogen cycling in terrestrial ecosystems, it deserves to explore how elevated nitrogen deposition affects soil N transformations in the N-rich soil of tropical forests. Overall, this manuscript was well written and easy to read, but the current version is suffering from some critical defects.

**Response:** Thanks for the positive evaluation to our work! We carefully revised the manuscript based on your suggestions. Our point-by-point responses to your comments are listed below. Hope you would find these revisions satisfactory.

***Reviewer comment:*** First, this study measured the net mineralization and nitrification, completely different from gross mineralization and nitrification. To this point, the title of this study is not appropriate, because net mineralization and nitrification actually include the balance of various transformation processes such as ammoniation and immobilization, which conceals real nitrogen transformation processes.

**Response:** Agreed. Net mineralization and nitrification rates essentially measures the net temporal changes in the pool size of inorganic N ($NO_3^-$ and $NH_4^+$) contents within the incubation period (in our case, 30 days). The limitation of field-assessed net rates can not disentangle the detailed gross transformation rates actually happening simultaneously. We therefore specified the N transformation rates as 'net N transformation rates' in the title and throughout the manuscript during this revision. In another study from our lab, Han et al. (2018) reported the responses of gross rates to N additions (Science of the Total Environment, 626: 1175-1187). We mentioned some of their results in our discussions.

***Reviewer comment:*** Second, the descriptions in Methods are not detailed and thus affect understanding of the results. For example, the descriptions about the specific time for nitrogen addition and sampling soil cores for net mineralization were unclear. Considering net mineralization is the difference of ammonium concentrations between 30 days, the time for nitrogen addition and the sampling of two soil cores is very important. If the sampling of second soil cores was just after nitrogen addition, mineralization could be overestimated because added N contributed to increase in soil ammonium concentrations.

**Response:** N additions were applied on the 24th of each month from September 2014 through October 2016. The incubations were carried out 9 times in September 2014, December 2014, March 2015, June 2015, September 2015, December 2015, March 2016, June 2016, and September 2016. Each incubation was started a couple of days before the N addition date and lasted for 30 days. We have provided these methodological details in the revised manuscript as the reviewer suggested.

*Reviewer comment:* Third, it is well known that nitrogen addition will lead to soil acidification. However, this study did not separate from inorganic nitrogen input from its acidification (also see Fig. 6). This strongly reduces the importance of this study, e.g. both low pH and higher inorganic nitrogen concentrations can show negative effects on nitrogen transformations.

**Response:** In the 2-year study period, we monitored the changes in both soil pH and inorganic N ($NH_4^+$-N and $NO_3^-$-N) contents after N additions and analyzed the relationship between these important factors and net N transformation rates. No significant relationship was found between them in the dry season (Table 1a). However, in the wet season, the net N transformation rates ($R_m$ and $R_n$) had significantly positive correlations with $NO_3^-$-N content, but had significantly negative relationships with soil pH and $NH_4^+$-N contents (Table 1b and Fig. 5). Since changes in pH actually was induced by the nitrogen additions, we were therefore not able to separate the N addition effects from the acidification effects with our experimental design (only N input was manipulated). Further studies manipulating both soil acidification and N addition at the same time might be helpful in teasing out the two kinds of effects.

*Reviewer comment:* Fourth, it is very good to include the measurements of N-related functional gene abundance, but it is a pity that N-related functional gene abundance was not related with the specific nitrogen transformation processes. As a result, it is difficult to make a microbial mechanism explanation for net mineralization and nitrification. Before the manuscript is accepted to publish, the above issues should be well clarified.

Response: Yes, we only found $N_2O$ emission exhibited a significantly relationship with *nosZ* gene abundance in this study. We did not find a significant relationship between the AOA abundance and net N mineralization rates. The main reason may be that net N mineralization rate actually measures the net temporal changes in inorganic N pool sizes, which are governed by several specific gross input and output rates such as gross mineralization and immobilization. It is possible that the functional genes abundance may have closer relationships with the gross N transformation rates. Some of such relationships have been reported in a recent study by Han et al. (2018) using soil samples taken from the same experimental plots as in this study. We have added these descriptions in the revised discussion to further explore the relationships between functional gene abundance and N transformation rates.

---

## Author Response (AR2)

Yanxia Nie and co-authors
Correspondence to: Weijun Shen (shenweij@scbg.ac.cn)

Thanks for the nice suggestions. The manuscript has been revised based on the suggestions. We hope that you would find the revision satisfactory.

*Response to Anonymous Referee*
*Reviewer comment: 1) A careful proofreading should be made to improve language, e.g., 'lab' should be 'laboratory'.*
**Response:** We carefully checked the language and corrected them as suggested (see P11L3-L11, P2L5, P7L13, and P9L1).

*Reviewer comment: (2) I suggest that the authors should add one sentence in the end of conclusion to clarify the future direction based on this study.*
**Response:** Thanks for your suggestion. We suggested that long-term continuous N enrichment should be undertaken to decipher the relationships between soil N transformation processes and microbial functional genes. Management strategies should also be developed in order to alleviate the negative effects of elevated N deposition on soil microbial functions. Please see P15L3-6.

*Response to Associate editor*
*Reviewer comment: (1) In the materials and methods, please add few sentences showing how the addition of N is related to wet N deposition*
**Response:** The ambient wet N deposition rate was about 34.6 kg N ha$^{-1}$ yr$^{-1}$. Our experimental N addition rate was 35 kg N ha$^{-1}$ yr$^{-1}$ for the LN treatment, 70 kg N ha$^{-1}$ yr$^{-1}$ for the MN treatment, and 105 kg N ha$^{-1}$ yr$^{-1}$ for the HN treatment. Therefore, our experimental N addition levels were 1, 2, and 3 times of the ambient wet N deposition rate for the LN, MN and HN treatments, respectively. We added the information in the materials and methods (see P5L17-18, and P5L24-25). Thanks for the suggestion!

*Reviewer comment: (2) maybe the title could be rephrased as: The simulated N deposition accelerates net N mineralization and nitrification in a tropical forest soil*
**Response:** Agreed. The title has been rephrased as suggested.

*Reviewer comment: (3) One conclusion sentence might be added at the end of abstract showing the implication and importance of N deposition on soil N turnover*
**Response:** Thanks! A conclusion sentence has been added at the end of the abstract by emphasizing the main findings of this study and future research needs (see P2L20-24)**.**

***Reviewer comment:*** *(4) The last paragraph of Introduction might be slightly rephrased to highlight the aim, i.e., the short-term N addition is to investigate the possible effect of N deposition on ecosystem.*

**Response:** Following your suggestion, we have rephrased the three research aims for clarity (see P4L32-35 and P5L1-5).

[revised manuscript text omitted]

Fig. 2

[Figure]

[Figure]

Fig. 3

Fig. 4

[Figure]

Fig. 5

[Figure]

Fig. 6

[Figure]